# A Novel Gelatin-Based Sustained-Release Molluscicide for Control of the Invasive Agricultural Pest and Disease Vector *Pomacea canaliculata*

**DOI:** 10.3390/molecules27134268

**Published:** 2022-07-02

**Authors:** Jie Wang, Yuntian Xing, Yang Dai, Yingnan Li, Wenyan Xiang, Jianrong Dai, Fei Xu

**Affiliations:** 1Ministry of Education Key Laboratory of Industrial Biotechnology, School of Biotechnology, Jiangnan University, 1800 Lihu Avenue, Wuxi 214122, China; sunflower1230@163.com (J.W.); xingyuntian@jipd.com (Y.X.); yingnan3400189@163.com (Y.L.); 2National Health Commission Key Laboratory of Parasitic Disease Control and Prevention, Jiangsu Provincial Key Laboratory on Parasite and Vector Control Technology, Jiangsu Institute of Parasitic Diseases, Wuxi 214064, China; 15951581011@163.com (Y.D.); xiangwenyan1214@163.com (W.X.); djr0008@163.com (J.D.)

**Keywords:** invasive species, biological material carrier for sustained drug, *Pomacea canaliculata* control, environmental safety

## Abstract

*Pomacea canaliculata*, one of the 100 most destructive invasive species in the world, and it is an important intermediate host of *Angiostrongylus cantonensis*. The molluscicides in current use are an effective method for controlling snails. However, most molluscicides have no slow-release effect and are toxic to nontarget organisms. Thus, these molluscicides cannot be used on a large scale to effectively act on snails. In this study, gelatin, a safe and nontoxic substance, was combined with sustained-release molluscicide and was found to reduce the toxicity of niclosamide to nontarget organisms. We assessed the effects of gelatin and molluscicide in controlling *P. canaliculata* snails and eggs. The results demonstrated that the niclosamide retention time with 1.0% and 1.5% gelatin sustained-release agents reached 20 days. Additionally, the mortality rate of *P. canaliculata* and their eggs increased as the concentration of the niclosamide sustained-release agents increased. The adult mortality rate of *P. canaliculata* reached 50% after the snails were exposed to gelatin with 0.1 mg/L niclosamide for 48 h. The hatching rate of *P. canaliculata* was only 28.5% of the normal group after the treatment was applied. The sustained-release molluscicide at this concentration was less toxic to zebrafish, which means that this molluscicide can increase the safety of niclosamide to control *P. canaliculata* in aquatic environments. In this study, we explored the safety of using niclosamide sustained-release agents with gelatin against *P. canaliculata*. The results suggest that gelatin is an ideal sustained-release agent that can provide a foundation for subsequent improvements in control of *P. canaliculata*.

## 1. Introduction

As an important intermediate host of *Angiostrongylus cantonensis* (a zoonotic parasitic nematode) [1], the invasive apple snail *Pomacea canaliculata* (*P. canaliculata*) is listed as one of the world’s 100 most harmful invasive species [2]. *Angiostrongylus cantonensis* can cause eosinophilic meningitis in humans, and the snail host increases the spread rate and outbreak range of the disease [3]. In addition, *P. canaliculata* has established natural populations in several southern provinces in China and is currently widely distributed and is spreading because of its strong adaptability, rapid growth, high fecundity, and lack of natural predators. This species eats many rice seedlings and consumes approximately one million hectares of rice in southern China each year. The estimated annual cost of controlling these snails using pesticides is approximately USD 300 million [4]. Therefore, *P. canaliculata* control will not only help to prevent invasion by this species and the transmission of *A. cantonensis* [5], but will also reduce the economic losses due to crop damage.

Current methods for controlling *P. analiculate* include biological, physical, and chemical techniques. Biological controls typically entail using ducks that prey on *P. analiculate*. However, ducks rarely actively prey on *P. analiculate* because it is difficult for ducks to digest the lectin-protected eggs of *P. analiculate* [6]. Physical controls typically involve manual harvesting, but this process involves significant labor and resources and is inefficient [7]. Chemical controls are the most popular method used for controlling snails due to their effectiveness. Currently, spirulina ethanolamine salt, metaldehyde, and niclosamide are currently used as chemical control agents [8]. However, the rapid spread of *P. analiculate* is making chemical controls less desirable. In addition, chemical agents can be toxic to nontarget organisms [9,10], produce pesticide residues [11], and may produce drug resistance [12]. Therefore, these chemicals cannot be used in large quantities. In addition, snail-killing agents have no slow-release effect and cannot effectively act on snails growing on aquatic plants, along the water surface, or during rainy weather. Moreover, most research has focused on controlling adult *P. analiculate*, while few studies have investigated the control of *P. canaliculata* eggs [13]. If adult *P. canaliculata* die, the remaining eggs can quickly restore the population after hatching. Therefore, effective control methods must target both adults and eggs, and slow-release drugs should be considered. A high-concentration calcium chloride salt solution has been reported to inhibit the hatching of snail eggs, but the application of this method is limited because it will salinize the land. Therefore, it is important to find effective methods for controlling these snails and their eggs.

In 1980, Pretice mixed 70% wettable molluscicide with flour and sand to produce different granule formulations, with the molluscicidal effect of these granules found to exceed 95% on marshland in Saint Lucia [14]. In China, Cai Dequan et al. [15] used niclosamide powder mixed with a carrier to prepare a granular controlled-release agent, which prolonged the molluscicidal effect and provided reduced toxicity to fish. Recently, chlorosalicylicamide sustained-release granules prepared by adding surfactant, adhesive, defoamer, and lubricant were also used to kill these snails [16]. The death rate of the snails exceeded 95% after 14 days of application. However, unlike snails that live in marshy and mountainous terraces, the granules could not be sprayed on *Pomacea canaliculata*, which live in ditches and deposit eggs near aquatic plants. Therefore, to protect the ecological environment, it is necessary to develop additional environmentally friendly sustained-release molluscicides. 

Hydrogel-based drug delivery systems have recently attracted the attention of the scientific community due to their superior properties. Hydrogels are soft polymer materials with a 3D structure [17] that have been used in several novel applications, including drug delivery [18,19], wound dressing [20,21,22], and tissue engineering [23,24], among other technical applications [25]. Hydrogels can promote the release of encapsulated drug molecules via environmental changes when the target drug is enclosed in a hydrogel matrix. In recent years, hydrogel materials and drug-loaded hydrogels have been used as antiparasitics due to their mechanical strength, high stability, large drug loading capacity, and long release time [26,27,28]. Hydrogels are typically made of natural [29] or synthetic polymers [30]. Gelatin is a natural polymer produced by collagen hydrolysis, containing 18 amino acids and 90% collagen, and it swells in water at 35–40 °C to form a thermally reversible gel.

In this study, gelatin, *P. canaliculata*, and *P. canaliculata* eggs were used to analyze gelatin’s performance as a biomaterial drug carrier and to determine the ability of gelatin containing a safe dose of niclosamide to kill adult snails and prevent their eggs from hatching. We also evaluated gelatin’s safety for zebrafish growth and performed a field efficacy test to assess the feasibility of using biological material carriers to prevent and control snails. This study will help researchers to identify environmentally friendly methods for controlling *P. canaliculata*.

## 2. Materials and Methods

### 2.1. Reagents and Equipment

Materials included the following: Niclosamide (CAS:50-65-7; purity ≥ 98%) was purchased from Sigma Aldrich (St. Louis, MO, USA). Sock solution was prepared in high performance liquid chromatography (HPLC) grade dimethyl sulfoxide (DMSO), gelatin (Aladdin), metaldehyde 60 GR (Guangdong Nongmi Biotechnology Co., Ltd., Sihui City, China), 50% niclosamide ethanolamine (Shanghai Hulian Biopharmaceutical Co., Ltd., Shanghai, China), CHCA (Brucker-Daltonics, Billerica, MA, USA), acetonitrile, trifluoroacetic acid (TFA), methanol (HPLC grade) (Sigma, USA), pure water (Millipore, Billerica, MA, USA), an electronic balance (Mettler Toledo Instruments Co., Ltd., Columbus, OH, USA), a constant-temperature water bath (Dongxing Building Materials Testing Equipment Co., Ltd., Shanghai, China), an ARES-G2 rheometer (TA Instruments, New Castle, DE, USA), a Hitachi SU1510 Scanning Electron Microscope (Tokyo, Japan), a FreeZone Plus 6 L freeze-dryer (Labconco, Kansas City, MO, USA), a microplate reader (Thermo, Waltham, MA, USA), a fluorescence confocal microscope (Leica, Buffalo Grove, IL, USA), and an Agilent 1260 high-performance liquid chromatograph (Agilent Technologies, Santa Clara, CA, USA).

### 2.2. P. canaliculata and Eggs

Eggs of *P. canaliculata* used in this study were collected from aquatic plants in Taihu Lake in Wuxi, Jiangsu (N 31°15′35.77″ and E 120°12′1.73″) and incubated in the laboratory for 8 weeks until they became snails. During the domestication period, the snails were fed as much lettuce as they could eat for 2 h once per day. Feeding ceased 24 h before the experiment started, and fecal matter and food residue were promptly removed. The snails were not fed during the experimental period. The room temperature was maintained at 27 °C, and the daily light lasted 12–14 h. Tap water dechlorinated with light for more than 24 h was used. The body length of the snails in the study was 14.5 ± 2.2 mm (body weight: 1.15 ± 0.23 g). 

In addition, we selected a matching batch of eggs with similar egg sizes on the stems of plants in Tai Lake to ensure comparability. The eggs were acclimated for 7 d under the same experimental conditions. 

### 2.3. Gel Formation Experiment

The gelatin was mixed with water to obtain 0.1%, 0.5%, 1.0%, 1.5%, 2.0%, and 3.0% gelatin systems. The gelatin–water mixed system was left at 25 °C for more than 1 h to allow the gelatin to fully swell to obtain the gelatin solution. The gelatin solution was dissolved at 40 °C. Next, 500 μL of 0.1–3.0% (*w*/*v*) gelatin solution was added to a 1.5 mL EP tube and redissolved in a 50 °C water bath. The solution was then maintained at 4 °C in a refrigerator. After the gel formed, the EP tube was removed and placed upside down. If the polymer molecules cross-linked to form a hydrogel, their mobility was restricted, and they remained at the bottom of the EP tube. Solutions that did not form a gel flowed down the tube wall. These observations were captured by a camera.

### 2.4. Microrheology Experiment 

The microrheology experiment was performed using an ARES-G2 (TA Instruments, New Castle, DE, USA) at room temperature with 25 mm parallel plates and a 5 mm gap. Gelatin gels (0.1–3.0% *w*/*v*) were prepared by treating gelatin (Aladdin, Shanghai, China) in 20 mM phosphate buffer (pH 7.4) containing 100 mM NaCl for 30 min at 37 °C. The gel was placed in the gaps of the parallel plates and frequency sweeps were performed in the 0.1–10 Hz range with a constant 1% strain; strain sweeps were performed in the 0.1–100% range with a constant oscillation frequency of 1 Hz. Young’s modulus was evaluated using a load meter, model-1605VCL (Aikoh Engineering Co, Ltd., Osaka, Japan). Gels were prepared in polymethylmethacrylate molds (ID/OD: 6/8mm) set on parafilm (Parafilm, Bemis Corporation, Neena, WI, USA), as described above. Young’s modulus G was calculated using the equation E = δσ/δε, where δσ is the stress and δε is the strain in the linear elastic region. Three separate measurements were performed for each sample [28]. 

### 2.5. Scanning Electron Microscopy (SEM)

Different concentrations of gelatin solutions were freeze-dried using a freeze-drying system. A thin layer of palladium/gold alloy was placed on the lyophilized hydrogel to improve the surface conductivity for SEM. Morphological and microstructural images of the gelatin were acquired using a Hitachi SU1510 SEM at an accelerating voltage of 5 kV.

### 2.6. Preparation of Niclosamide Gelatin and Determination of Sustained Release and Residues 

The 1.0% and 1.5% gelatin systems were prepared according to the above method. The gelatin solution was redissolved at 40 °C, and niclosamide was added to form a gelatin–niclosamide–water mixed solution at a concentration of 1 mg/L. The gelatin–niclosamide–water mixed solution was cooled in 24-well plates (1 mL per hole) to form a gel for use as a sustained-release molluscicide.

After the gelatin had formed, ddH_2_O (500 μL) was added to each well. The contents of niclosamide and gelatin in the aqueous solutions were determined at 30 min and 1, 2, 4, 6, 8, 12, 24, 36, 48, 72, 96, 144, 168, 192, 216, 240, 360, and 480 h. Next, 500 μL acetonitrile was mixed into the solutions with gelatin. The supernatant was obtained via centrifugation, which was detected by high-performance liquid chromatography (HPLC). The HPLC parameters were as follows. An AgilentZORBAX SB-C18 column was equipped with an Agilent 1260 system. The mobile phase (MP) was methanol: water = 90:10 (V/V), pH = 4, filtered using a 0.22 mm porous membrane at a column temperature of 25 °C (room temperature). Peak detection and analysis were conducted at 310 nm. The MP flow rate was 1.0 mL/min and the injection volume was 5 μL. The samples with niclosamide were recognized according to the standard retention time. On the other hand, the concentrations of niclosamide were calculated using a standard external method.

### 2.7. Activity Determination of Niclosamide against P. canaliculata

The mass concentrations of niclosamide were 0, 0.05, 0.1, 0.5, and 1 mg/L, and 30 mL of gelatin with each concentration was placed in a 50 mL Petri dish. *P. canaliculata* specimens with suitable growth and complete shells were selected and placed in Petri dishes of each concentration for treatment. Ten snails were placed in each dish. Three replicates were set for each concentration, along with a water control. The Petri dishes were covered to ensure ventilation while preventing the snail eggs from escaping after hatching. The mortality of *P. canaliculata* in each group was recorded at 48, 72, and 96 h after treatment. The determination method and standard for the death of *P. canaliculata* were as follows: The *P. canaliculata* specimens were moved to a glass containing nonchlorinated water and covered with a breathable wire mesh to ensure ventilation and prevent live *P. canaliculata* snails from escaping. If a *P. canaliculata* snail did not crawl against the glass wall, we opened the shell and touched the snail. If it did not shrink, we assumed it was dead.

### 2.8. Activity Determination of Niclosamide-Contained Gelatin against P. canaliculata Eggs

For the next part of the experiment, 30 mL measures of gelatin solution containing niclosamide at mass concentrations of 0, 0.05, 0.1, 0.5, and 1 mg/L were prepared and placed in 50 mL Petri dishes. We selected well-grown snail eggs in complete masses and placed them in Petri dishes of each concentration for processing, with one mass per dish. Three replicates were set for each concentration, and water and niclosamide were used as separate controls. The Petri dishes were covered with a ventilating cover. The mortality rates of the egg masses of *P. canaliculata* in each group were recorded at 5, 10, and 15 d after treatment.

### 2.9. Toxicity Assay of Chlorinated Niclosamide against Zebrafish

Zebrafish (male, aged 8 weeks, and purchased from the Wuxi Flower and Bird Market) were acclimated in a 25 °C rearing box for one week before use in the experiment. The average zebrafish weight was 0.68 ± 0.07 g, and the average length was 38.5 ± 0.3 mm. Feeding ceased 24 h before the experiment, and the fish were not fed during the experiment. Zebrafish selected for the experiment were strong, responsive, and of the same size.

Nine 500 mL glass beakers were placed in an incubator under constant temperature and humidity at 25 °C, and 50 mL gels containing 0, 0.05, 0.1, 0.5, and 1 mg/L niclosamide were put into five of the glass beakers. The other four beakers were used as positive control groups with niclosamide solutions of 0.05, 0.1, 0.5, and 1 mg/L. Ten zebrafish were selected and placed in each beaker. During the experiment, the photoperiod was 14 h light and 10 h darkness. The number of dead fish was observed and recorded every day. Three replicates were set for each concentration, from which an average was calculated.

### 2.10. Data Statistics

Statistical significance analysis was performed using the SPSS 20.0 software. The Probit method in SPSS20.0 was used to analyze the LC_50_ (50% lethal concentration) of niclosamide gelatin sustained-release agent for *P**. canaliculata*. The SPSS20.0 software’s Spearman rank correlation coefficient method was used to statistically analyze the correlation between the niclosamide gelatin sustained-release agent and the mortality of *P**. canaliculata*. *p*-value of <0.05 was defined as significant.

## 3. Results and Analysis

### 3.1. Gelatin Inversion Experiment

Gelatin solutions at 0.1%, 0.5%, 1.0%, 1.5%, 2.0%, and 3.0% (*w*/*v*) were prepared and placed at room temperature (25 °C) and under freezing conditions (4 °C) for 30 min to observe gel formation with different gelatin concentrations. Gelatin did not form at gelatin concentrations of 0.1% and 0.5%. Stable hydrogels formed when the gelatin concentration reached 1.0% and above (Figure 1). Gelatins of different concentrations were maintained as solutions when placed in an incubator at 40 °C, demonstrating that gelatin forms a gel at room temperature and dissolves into a solution at temperatures higher than the T_m_ value. This result indicates that gel formation is reversible.

### 3.2. Rheological Test

To further confirm gel formation and quantify the hydrogel strength, microrheology was used to measure the storage modulus (G′) and loss modulus (G″) values of the gels at different concentrations at room temperature. The results showed that the G′ of the 0.1% gelatin solution was lower than the G″, while the G′ value was significantly lower than those of the gelatin solutions with other concentrations (Figure 2), indicating that this solution did not form a hydrogel. The G′ and G″ values were observed to be similar for the 0.5% gelatin solution, forming a liquid gel. When the gelatin concentration reached 1.0% and above, the G′ values of all solutions were significantly higher than the values of G″ at room temperature, indicating that the solutions formed hydrogels. The modulus of elasticity for 1.0%, 1.5%, 2.0%, and 3.0% gelatin reached 87.6 ± 0.99, 169.7 ± 3.85, 773.8 ± 103.44, and 1352.3 ± 186.59 kPa (mean ± S.E.), respectively. Thus, the gelatin concentration was positively correlated with hardness. Gelatin hardness increased as the concentration increased, and the lowest concentration at which gelatin was formed was determined to be 0.5% (*w*/*v*). The results of the rheological analysis were consistent with the results of the test tube inversion test. 

### 3.3. Microstructures of the Gelatin

As shown in Figure 3, collagen fibers packed with loose macroporous structures were observed on the gelatin surface at concentrations of 0.1% and 0.5% (Figure 3A–F). Relatively dense mesoporous structures were observed on the gelatin surface at concentrations of 1.0% and 1.5% (Figure 3G–L). A fibrous pore structure was observed on the gelatin surface at concentrations of 2.0% and 3.0% (Figure 3M–R). These results indicate that the lower the gelatin concentration, the looser the pores; conversely, the higher the gelatin concentration, the denser the pores. A previous study showed that the drug release rate is related to microstructures; the drug release rate in more compact structures with higher mechanical strength was found to be lower than that in looser and weaker structures with larger pore sizes [31]. Since the 0.1% and 0.5% gelatin solutions were soft and did not form solid states, while the 2% and 3% gelatin solutions formed hard solid states, concentrations of 1.0 and 1.5% were selected for the drug-release gelatin, and the drug release was compared between the two.

### 3.4. Drug Release and Residue 

To further observe the sustained release and residues associated with different concentrations of gelatin, we used 1mg/L niclosamide encased in 1.0% and 1.5% gelatin and observed the release and residue of niclosamide. The 1.0% and 1.5% gelatin formulations released, respectively, 5.25 ± 0.24% and 6.27 ± 0.18% of niclosamide after 30 min in the release buffer, 24.15 ± 0.78% and 35.08 ± 1.20% of niclosamide after 72 h, and 42.52 ± 2.67% and 51.54 ± 1.26% of niclosamide after 240 h; the drug release and residue remained at stable levels throughout the process (Figure 4). These results demonstrate that 1.0% and 1.5% gelatin have sustained-release abilities, and both can effectively release niclosamide. However, 20–30% niclosamide remained in the gelatin until 480 h. We also observed that the release rates differed depending on gelatin concentrations. Compared to 1.0% gelatin, 1.5% gelatin featured a lower drug release rate but a longer drug retention time in the gel (reaching 20 d). Thus, the gel formed with 1.5% gelatin was more stable and also showed resistance to decreasing the gelatin surface due to colloidal swelling in aqueous solution, giving it significant application potential. 

### 3.5. Test of Control Effect of Niclosamide-Containing Sustained-Release Gelatin Formulation on P. canaliculata (LC_50_)

To observe the control effects of niclosamide-containing sustained-release gelatin preparations on *P. canaliculata*, 1.5% sustained-release gelatin with final niclosamide concentrations of 0, 0.05, 0.1, 0.5, and 1 mg/L were prepared. The control effects and LC_50_ values of these preparations were observed 12, 24, 48, 72, and 96 h after the preparations were applied to *P. canaliculata*. The results demonstrated a significant correlation between snail mortality and niclosamide concentration at the same temperature (Figure 5a). The LC_50_ values of the niclosamide sustained-release gelatin formulation for snails at 12, 24, 48, 72, and 96 h were 0.907, 0.589, 0.156, 0.05, and 0.025 mg/L, respectively (Figure 5b). *p*-values were less than 0.05 at 12, 24, 48, and 72 h. All data are displayed in Appendix A.

### 3.6. Test of Control Effect of Niclosamide-Containing Gelatin on P. canaliculata Eggs

To observe the control effect of niclosamide-containing gelatin on snail eggs, gelatin containing final concentrations of 0, 0.05, 0.1, 0.5, and 1 mg/L niclosamide was prepared, and *P. canaliculata* eggs were observed after these gelatin treatments were applied to the eggs for 5, 10, and 15 d (Figure 6). *P. canaliculata* eggs without the drug or gelatin were used as the negative control, and 0.1 mg/L niclosamide solution was used as the positive control. The average hatching rates of the eggs without niclosamide and gelatin at 5, 10, and 15 d were 17.3 ± 0.92%, 32.67 ± 0.27%, and 60.23 ± 3.98%, respectively. Gelatin containing 0 mg/L niclosamide barely affected the hatching rate. There was a slight decrease in the hatching rate to 13.37 ± 0.57%, 29.13 ± 1.85%, and 41.43 ± 1.88% when the gelatin contained 0.05 mg/L niclosamide. When the niclosamide concentration increased to 0.1 mg/L, the hatching rate decreased to 4.21 ± 0.57%, 8.57 ± 1.04%, and 17.07 ± 2.08% after 5, 10, and 15 d of treatment, respectively. The hatching rates were 0%, 4.28 ± 1.39%, and 13.13 ± 0.83% after the application of 0.5 mg/L niclosamide gelatin for 5, 10, and 15 d, respectively. The incubation rate of gelatin containing 1 mg/L niclosamide was 0% at 5 d, 2.35 ± 0.77% at 10 d, and 6.57 ± 0.52% at 15 d. Most eggs did not hatch. As the drug concentration and application time increased, niclosamide-containing gelatin effectively prolonged the incubation time of eggs and reduced total egg production (Figure 6 and Appendix A).

### 3.7. Acute Toxicity Test of Niclosamide-Containing Gelatin on Zebrafish

To identify concentrations of the sustained-release agent suitable for use in aquatic environments (to avoid affecting fish and other organisms), we analyzed the toxicity of drug-containing gelatin sustained-release agents on zebrafish (Figure 7). When exposed to high concentrations (0.5 and 1 mg/L) of niclosamide solution, all the zebrafish died within 20 min; at a medium concentration (0.1 mg/L) of niclosamide solution, the fish began to die after 8 h and all died after 48 h; at low concentrations (0.05 mg/L) of niclosamide solution, the fish began to die at 48 h and the mortality rate was 50% after 96 h. However, under high concentrations of 0.5 and 1 mg/L of niclosamide sustained-release gelatin, only one or two zebrafish lost the ability to swim at 72 h; 10% and 20% of zebrafish died at 96 h. Only 13.3% and 23.3% of zebrafish died from 120 to 192 h. No zebrafish died when the niclosamide concentrations were 0.05 and 0.1 mg/L, and no zebrafish died in pure gelatin without niclosamide. These results indicate that at concentrations of 0.1 mg/L and below, the niclosamide gelatin sustained-release agent did not affect zebrafish in the aquatic environment. This concentration is optimal for use in sustained-release agents. All data are displayed in Appendix A.

## 4. Discussion 

The rapid reproduction of *P. canaliculata* is harming the ecology of wetlands and the Taihu Lake Basin and increases the risk of parasitic disease transmission. At present, the most commonly used biological, physical, and chemical methods cannot effectively control and prevent *P. canaliculata*.

Most existing slow-release molluscicide agents are produced using synthetic polymers or biochar as a carrier. However, synthetic polymers are often difficult to degrade and can easily cause environmental hazards. Additionally, biochar can damage the ecological beauty and aesthetic value of freshwater regions. Current technologies for preventing and controlling snail eggs involve use of a high-concentration calcium chloride salt solution to inhibit egg hatching. However, the field use of this method is greatly limited because calcium chloride increases the salinity of the land. Dai et al. developed a suspension agent containing niclosamide. As a low-toxicity and safe chemical drug that can control snails, niclosamide has molluscicidal properties but is also toxic to fish [32], limiting its use in aquatic environments. Therefore, it is important to research ecofriendly and effective methods for controlling *P. canaliculata*. 

To effectively control *P. canaliculata* snails and their eggs, the use of sustained-release drugs must be considered. In this study, a gelatin-based niclosamide sustained-release agent was developed based on the properties of different gelatin concentrations and sustained-release drugs. We also studied the ability of niclosamide gelatin to control *P. canaliculata* and their eggs. As the niclosamide sustained-release concentration increased, the mortality of *P. canaliculata* adults and eggs significantly increased. The egg hatching rate significantly decreased after application of the sustained-release agent.

To select the appropriate gelatin concentration, we observed the gel-forming properties and microstructures of different gelatin concentrations through inversion experiments, rheological experiments, and SEM experiments. Gelatin at a concentration of 1.0–1.5% (*w*/*v*) offered sufficient gel-forming properties, was able to form a stable temperature-controlled reversible hydrogel, and had moderate pores, all features which make gelatin effective at encapsulating and releasing drugs. We also observed the sustained-release ability of gelatin drugs using high-performance liquid chromatography. We found that 1.5% gelatin had more trouble absorbing water compared to 1.0% gelatin, leading to a slower rate of release and a longer retention time (up to 20 d). Therefore, gelatin is also an ideal sustained-release agent and can effectively prevent photolysis from inactivating niclosamide within 7–10 d. Moreover, this slow drug release reduces the harm that niclosamide can cause to the environment and more effectively impacts the whole egg growth and incubation cycle, which is generally 15–20 d.

We observed the control effect of niclosamide-containing gelatin on *P. canaliculata* adults and eggs at concentrations of 0.05, 0.1, 0.5, and 1 mg/L for 12–96 h. The hatching rates of the eggs at these gelatin concentrations were assessed after 5–15 d. There was a significant correlation between mortality and niclosamide concentration. As the niclosamide gelatin concentration increased, the time for snail formation to reach the LC_50_ value gradually decreased. After 48 and 72 h of niclosamide gelatin sustained-release action on *P. canaliculata*, the LC_50_ was found to be 0.088 and 0.033 mg/L, respectively. Additionally, a higher concentration of niclosamide gelatin and longer action time corresponded to a lower hatching rate compared to the control group and the niclosamide-free gelatin group. The hatching rate associated with gelatin containing 0.1, 0.5, and 1 mg/L niclosamide after 10–15 d of treatment was, respectively, 26.2–28.5%, 13.1–21.9%, and 7.2–11.0% that of the normal group. Most eggs did not hatch and many of the hatched young snails remained on the gel surface, leading to a low survival rate.

Zebrafish are highly sensitive to niclosamide. To comprehensively assess the risks associated with application of sustained-release solutions, we conducted an acute toxicity test on zebrafish using niclosamide-containing gelatin. We found that the niclosamide gelatin sustained-release formulation was less toxic to zebrafish than niclosamide solution. When treated with high concentrations (0.5 and 1 mg/L niclosamide gelatin sustained-release treatment), the zebrafish mortality was only 1.0–2.0% after 4 days, while all zebrafish died within 20 min under the same concentration of niclosamide solution. When treated with 0.05 and 0.1 mg/L niclosamide gelatin sustained-release agent, the zebrafish did not die over the course of 8 d. Under the same concentration of niclosamide solution, zebrafish began to die within 12–24 h, and all died within 96 h. 

Moreover, we found that the toxicity of niclosamide to fish did not increase with the accumulation and release of niclosamide after 120 days. The half-life of niclosamide in direct sunlight was found to be 4.7 days [33]. After 4.7 days, niclosamide began to degrade gradually in water and had no effect on the accumulation of toxicity in fish. These results indicate that the 0.1 mg/L niclosamide sustained-release preparation can be used to control *P. canaliculata* and will not be toxic to fish in the surrounding aquatic environments.

*P. canaliculata* typically lives in aquatic environments and lays eggs on the stems and leaves of nearby plants to reproduce. This species lays hundreds of calcareous eggs and is widespread in paddy fields and other wetlands in China [34]. It is difficult for common molluscicides to penetrate and kill the whole egg mass, while chemical treatments flow to the bottoms of plant stems and leaves, thereby contaminating water and exerting toxic effects on fish [32]. Gelatin is a natural hydrogel with gel-forming, drug encapsulation, and sustained-release properties. We demonstrated that gelatin is not toxic to fish and can effectively control *P. canaliculata* adults and eggs without affecting the surrounding environment. Therefore, gelatin is a natural, safe, and nontoxic slow-release agent that can be used to control *P. canaliculata*.

Our results confirmed that natural gelatin is a better drug release agent. 0.1 mg/L niclosamide gelatin sustained-release agent showed high molluscicidal activity and low toxicity to fish. Some researchers have studied the ability of natural products that are not harmful to the environment to effectively prevent and control breeding around bodies of water and thus control *P. canaliculata* (e.g., saponin), which also provides us with the basis study of new sustained-release preparations for the future. However, the efficacy and the ecotoxicological impact of sustained-release niclosamide in the field needs further determination. This study explored the ability of a gelatin-based sustained-release niclosamide solution to control *P. canaliculata* while reducing environmental impacts, thereby laying a foundation for future improvements to sustained-release molluscicide solutions.

## 5. Conclusions

In this study, a gelatin-based sustained-release molluscicide with high efficiency and low toxicity was developed without environmental harm. Our results demonstrated that gelatin with a mass volume concentration of 1.5% (*w*/*v*) could form a stable hydrogel with a typical mesh structure, which makes it a good carrier for sustained drug release. Moreover, the niclosamide drug can effectively act on *P. canaliculata*. The gelatin overcomes the shortcomings of existing molluscicides, such as high toxicity, dilution in rainwater, high application concentrations, the inability to continuously kill *P. canaliculata*, and widespread environmental problems. In addition, niclosamide gelatin has a sustained-release effect, and the drug in the gelatin can maintain the synergistic effects needed to effectively control both snails and eggs. The mortality rates after applying 0.1 mg/L molluscicide gel for 48 h and 3 d were >50% and 100%, respectively, and the hatching rate of eggs was only 28.5% of the normal rate. In this work, we studied a novel gelatin-based sustained-release molluscicide to control *Pomacea canaliculata* that does not harm nontarget organisms such as fish in aquatic environments.

## Figures and Tables

**Figure 1 molecules-27-04268-f001:**
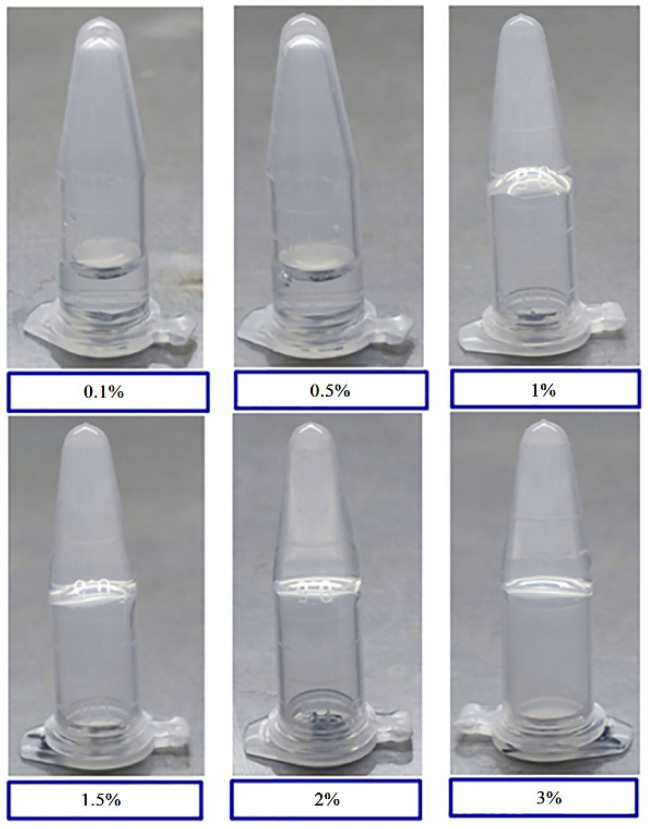
Tube inversion test of hydrogel formation with various gelatin concentrations. Vials containing 0.1%, 0.5%, 1%, 1.5%, 2%, and 3% (*w*/*v*) gelatin were subjected to a temperature of 4 °C for 30 min and inverted for image collection.

**Figure 2 molecules-27-04268-f002:**
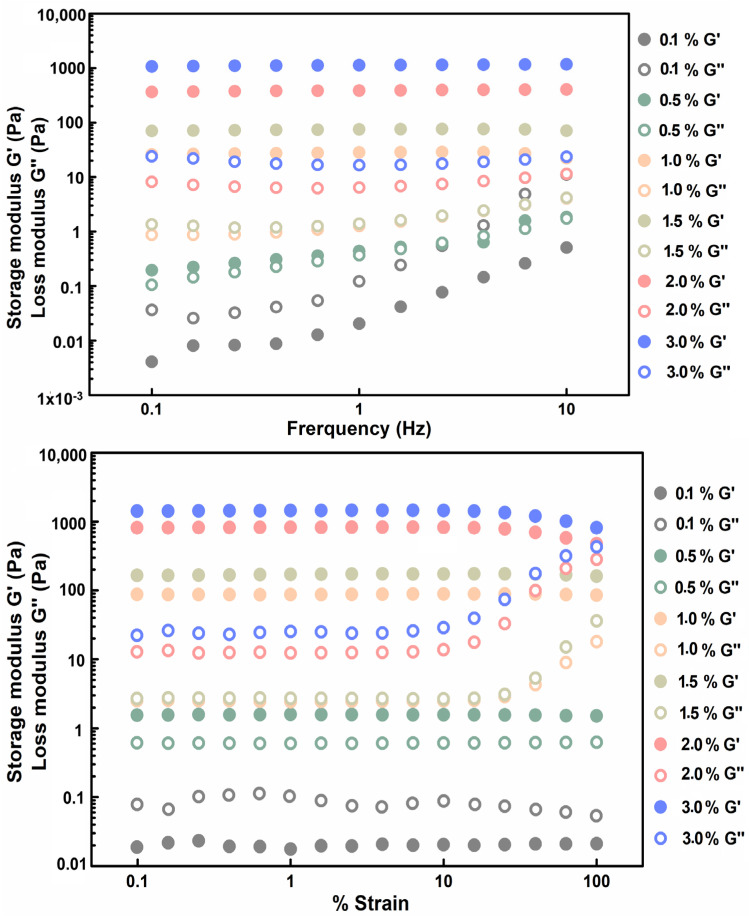
Microrheology results of different concentrations of gelatin (*w*/*v*). Storage and loss moduli, G′ and G″, of the 3.0% gelatin (blue), 2.0% gelatin (rose), 1.5% gelatin (yellow), 1.0% gelatin (pink), 0.5% gelatin (green), and 0.1% gelatin (grey).

**Figure 3 molecules-27-04268-f003:**
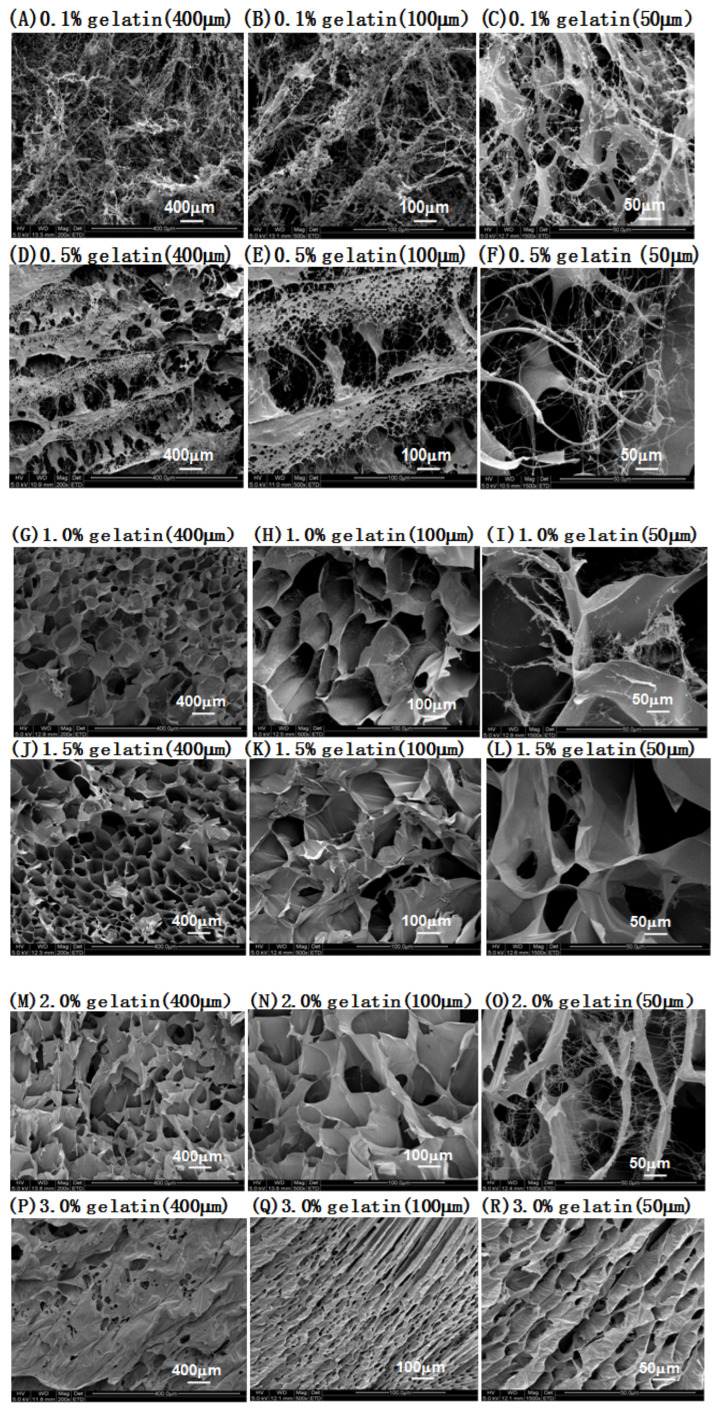
SEM images of sustained-release preparations based on different gelatin concentrations. (**A**–**D**) SEM images of sustained-release preparations based on different gelatin concentrations. After lyophilization, the microstructures of the gel were cut into longitudinal sections and visualized using scanning electron microscopy (SEM). The SEM of (**A**–**F**) 0.1% and 0.5% gelatin (longitudinal), (**G**–**L**) 1.0% and 1.5% gelatin (longitudinal), and (**M**–**R**) 2.0% and 3.0% gelatin (longitudinal) preparations are shown. Scale bar = 400 μm for A, D, G, J, M, and P; 100 μm for B, E, H, K, N, and Q; and 50 μm for C, F, I, L, O, and R.

**Figure 4 molecules-27-04268-f004:**
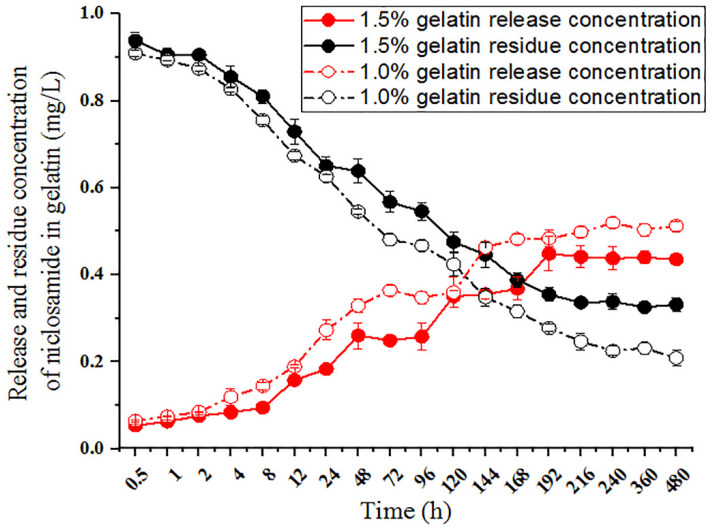
Release and residue of niclosamide in the gelatin preparations. Solutions of 1 mg/L niclosamide and 1.0% and 1.5% (*w*/*v*) gelatin were mixed to form hydrogels at 4 °C on a 96-well plate. Niclosamide release was initiated by dropping water onto the gelatin surface. Niclosamide residue was initiated by dropping acetonitrile onto the gelatin.

**Figure 5 molecules-27-04268-f005:**
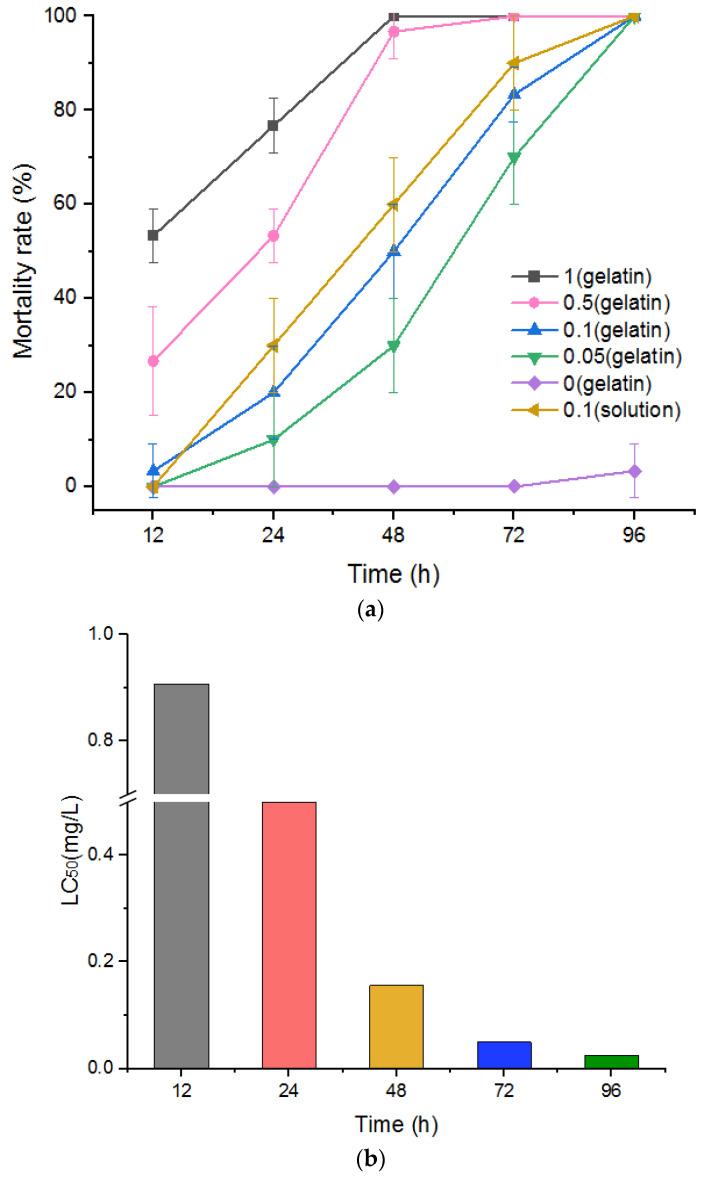
(**a**) Fitting curve of *P. canaliculata* mortality at different times and under different concentrations of sustained-release niclosamide gelatin preparation. Significantly different from the corresponding control, *p* < 0.05 (n = 3). (**b**) LC_50_ (mg/L) of *P. canaliculata* treated with niclosamide gelatin sustained-release agent at different concentrations.

**Figure 6 molecules-27-04268-f006:**
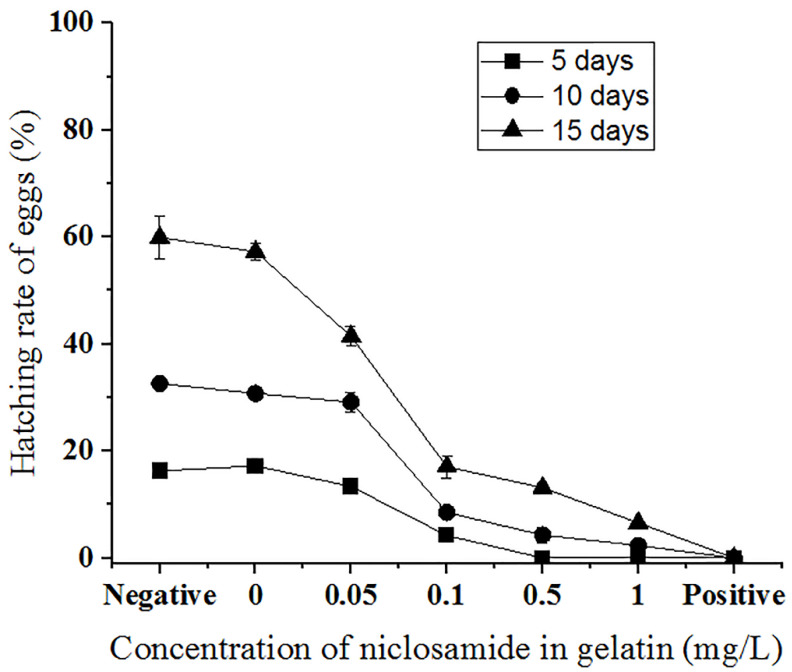
Hatching rate (%) of *P. canaliculata* eggs treated with different concentrations of niclosamide in gelatin for 5 days (square), 10 days (circle), and 15 days (triangle) under 1 mg/L niclosamide in 1.5% (*w*/*v*) gelatin. Negative: *P. canaliculata* eggs hatched in a culture dish without the gelatin or drug. Positive: eggs hatched in a culture dish with 0.1 mg/L niclosamide solution.

**Figure 7 molecules-27-04268-f007:**
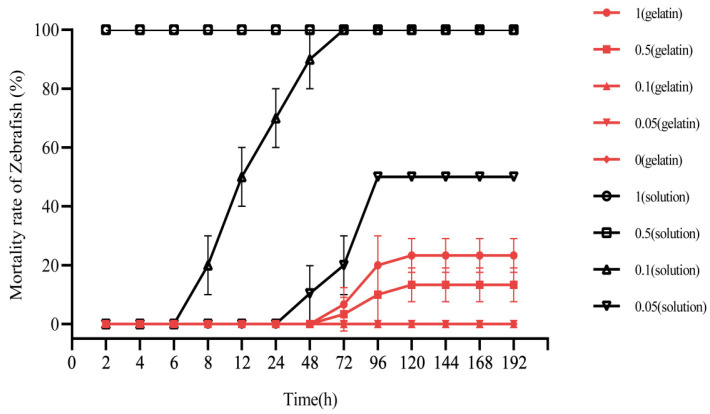
Mortality rates of zebrafish exposed to niclosamide under different concentrations of niclosamide gelatin or solution (%) (red line). Zebrafish exposed to niclosamide under 1 mg/L niclosamide gelatin (circle), 0.5 mg/L niclosamide gelatin (square), 0.1 mg/L niclosamide gelatin (equilateral triangle), 0.05 mg/L niclosamide gelatin (inverted triangle), and gelatin without niclosamide (diamond). Zebrafish (black line) exposed to niclosamide under 1 mg/L niclosamide solution (circle), 0.5 mg/L niclosamide solution (square), 0.1 mg/L niclosamide solution (equilateral triangle), and 0.05 mg/L niclosamide solution (inverted triangle). Three red lines of 0.1, 0.05 and 0 mg/L niclosamide gelatin are overlapped because the mortality rates of zebrafish were the same.

## Data Availability

The data presented in this work are available in the article and Appendix A.

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
