# Peer review of "A Novel Gelatin-Based Sustained-Release Molluscicide for Control of the Invasive Agricultural Pest and Disease Vector *Pomacea canaliculata"

_molecules, 2022, doi:10.3390/molecules27134268_

Round 1

Reviewer 1 Report

This study is well convinced, well conducted and excellently presented. The present study evaluated that the niclosamide retention time in 1.0% and 1.5% gelatin sustained-release agent can reach 20 days, and that the mortality rate of P. canaliculate and their eggs increases as the concentration of the niclosamide sustained-release agent increases.

The gelatin overcomes the shortcomings of the existing molluscicide, such as high toxicity, easily being affected by rainwater dilution, high application concentration, inability to continuously kill P. canaliculates, and widespread environmental problems. The results suggest gelatin is an ideal sustained-release agent, 26 which provides a foundation for subsequent improvements of way for P. canaliculata control with high efficiency and low toxicity without environmental harm.  I think that the manuscript is able to be published as the study evaluated that a novel gelatin-based sustained-release molluscicide is good option to control of Pomacea canaliculata, which does not harm non-target organisms such as fish in the aquatic environment.

I have only one question  why did author not observe mortality data after 24hour of application to identify concentrations of the sustained-release agent suitable in aquatic environments?

Author Response

Reviewer#1

Comments:
1.I have only one question why did author not observe mortality data after 24hour of application to identify concentrations of the sustained-release agent suitable in aquatic environments?

Author response: Thank you for allowing us to clarify. Please allow me address this concern from two aspects. First, snails would be needed in large numbers if the gelatin concentration were directly screened through the aquatic environment. In view of animal ethical issues, animal experiments involving large numbers of subjects have not been carried out for concentration detection and screening. Second, considering that there are many factors affecting bioassays, the test results would be less stable than those of instrumental analysis. Therefore, we used instrumental analysis to first observe the release ability of the drug in the gelatin and its molluscicidal effect to correctly screen the gelatin formulation.

Reviewer 2 Report

The manuscript entitled “A Novel gelatin-based sustained-release molluscicide for control of the invasive agricultural pest and disease vector, Pomacea canaliculata” analyzed the toxicity of niclosamide sustained-release gelatin in P. canaliculata (adults and eggs) and zebrafish (adults). The MS is current, well written and relevant. However, some points should be better clarified, such as the toxicity of niclosamide sustained-release gelatin to newly-hatched snails and its sublethal effects on zebrafish. Some additional concerns are detailed.

The keywords should not contain any words already in the title.

Line 67, 68. Oncomelania hupensis (use italics)

Line 68. Oncomelania hupensis = O. hupensis

Line 70. Pomacea canaliculate = P. canaliculate

Line 84. Delete “gelatin,”

Line 110. Confirm photoperiod

Line 163, 177. Describe the ventilating cover

Line 192-194. Describe the statistical analysis

Figure 5-1. Add x-axis legend

Figure 5-1. Add the lower and upper limits.

Add the acute toxicity test with newly-hatched snails.

Line 332-335; 402-403; 436-438. I recommend rewriting the text. Additional studies are needed to understand the sublethal effects of niclosamide sustained-release gelatin on zebrafish.

I recommend the analysis of the toxicity of niclosamide sustained-release gelatin in zebrafish embryos and larvae.

Conclusion. Future studies on the ecotoxicological impact of niclosamide sustained-release gelatin at different trophic levels are needed.

Reviewer 3 Report

The work by Wang J and colleagues is a study about the application of gelatin-based sustained-release of niclosamide that can be safely used to control the spread of P. canaliculata. However, the work is confused and lacking in the experimental part. In particular, the controls are missing or not correctly reported as well as the statistical tests used. This work should be considered as preliminary and it cannot be considered a regular paper.

Here below I attach my comments for the authors:

  • Methods, (L102-111): what kind of animal selection did you perfomerd? What is the female:male ratio? How old are the animals used in the experiments?
  • Methods, (L164-165): authors declared that snail were not fed during the experimental period (L108-109). Could the starvation affect the mortality of snail during the treatment? Control group data must be shown and a figure must be reported for snail similar to eggs one (Scheme 1). Moreover, Pomacea canaliculata is a freshwater snail that live in wet environment, did you provide a wet environment during the experimental period? Were Pomacea animals placed on gelatin gels?
  • Methods, (L-179-191): The authors compared the mortality of zebrafish exposed to niclosamide dissolved in water or in gelatin concluding that gelatin-based sustained release were less toxic compared the equal amount of dissolved niclosamide. However, as shown in Figure 7, the toxic effect may be delayed due to the progressive release of the compound whose effects can be the same of dissolved ones. Moreover, the dissolved concentrations of niclosamide released in solution from the gel will be different from those dissolved directly in water which will be the maximum. Therefore, the comparison should be performed with the same concentration of niclosamide in the water phase. The experiment must be repeated and improved with the proper concentrations. Moreover, points in figure 7 are indistinguishable (,g., 0.05(gelatin), 0.1(gelatin) and gelatin without figure 7).
  • Results, (L230-237): the authors have superficially described the results of SEM which are superfluous and it is not explained why the 1.0-1.5% gelatin percentage is the best for drug delivery.
  • Results, (L284—302): how did you perform the count of hatch? canaliculata eggs’clusters as you show in Scheme 1 have different size and this can affect drug diffusion, How did you select the eggs clusters?
  • Results, (L272): authors reported that there is a “significant linear positive correlation”, what kind of statistical test was applied?
  • The references are poorly written with names and surnames abbreviated incorrectly, watch the guidelines for authors.

Reviewer 4 Report

The article "A Novel gelatin-based sustained-release molluscicide for control of the invasive agricultural pest and disease vector, Pomacea canaliculata" is devoted to the actual problem of molluscicides development for the snails inhibiting in the environment. 

First principal question: in abstract authors are writing: The sustained-release molluscicide at this concentration is less toxic to zebrafish, which means that it can be safely used to control P. canaliculate in aquatic environments. - Could you please approve the fact that if the developed molluscicide is not affecting zebrafish - it is absolutely safety for the aquatic environments? Does only the zebrafish test is enough to identify the safety of such complex preparation use? Which kind of safety using tests are using in the internationally approved organization for this technique safety estimation? 

English must be improved, because in few places it is not scientific. For example, conclusions, line 425 - We developed - change to It was developed. Lines 437-438 - rephrase the sentense.

Whole impressions from the article - it is well-organized, experiment set up is correct, data presentation is of high quality.

The article can be accepted for publication

Author Response

Comments to the Author

The article "A Novel gelatin-based sustained-release molluscicide for control of the invasive agricultural pest and disease vector, Pomacea canaliculata" is devoted to the actual problem of molluscicides development for the snails inhibiting in the environment. Whole impressions from the article - it is well-organized, experiment set up is correct, data presentation is of high quality. The article can be accepted for publication

Comments:
1. First principal question: in abstract authors are writing: The sustained-release molluscicide at this concentration is less toxic to zebrafish, which means that it can be safely used to control P. canaliculata in aquatic environments. - Could you please approve the fact that if the developed molluscicide is not affecting zebrafish - it is absolutely safety for the aquatic environments? Does only the zebrafish test is enough to identify the safety of such complex preparation use? Which kind of safety using tests are using in the internationally approved organization for this technique safety estimation?

2.English must be improved, because in few places it is not scientific. For example, conclusions, line 425 - We developed - change to It was developed. Lines 437-438 - rephrase the sentense.

Author response: Thank you for bringing this up for us. Regarding language, we have revised the mentioned sentences and got the whole manuscript being polished by an English native speaker.
First of all, conclusions, line 425 the sentence “We developed a gelatin-based sustained-release molluscicide with high efficiency and low toxicity without environmental harm” has been changed to be “In this study, a gelatin-based sustained-release molluscicide with high efficiency and low toxicity was developed without environmental harm.” on page 15.

Second, conclusions, lines 437-438 the sentence “The article study on a novel gelatin-based sustained-release molluscicide for control of Pomacea canaliculata, which does not harm non-target organisms such as fish in the aquatic environment” has been rephrased to be “In this work, we studied a novel gelatin-based sustained-release molluscicide to control Pomacea canaliculata that does not harm nontarget organisms such as fish in aquatic environments.” on page 15.

Finally, we have invited MDPI magazine to carry out English Editing and got the whole manuscript being polished by an English native speaker. (MDPI English editing ID: English-44925 with certificate)
